# Epigenetic Modifications: An Unexplored Facet of Exogenous RNA Application in Plants

**DOI:** 10.3390/plants9060673

**Published:** 2020-05-26

**Authors:** Athanasios Dalakouras, Kalliope K. Papadopoulou

**Affiliations:** 1Department of Biochemistry & Biotechnology, University of Thessaly, 41500 Larissa, Greece; popypapad@gmail.com; 2Institute of Plant Breeding and Genetic Resources ELGO-DEMETER, 57001 Thessaloniki, Greece

**Keywords:** exogenous RNAi, dsRNAs, siRNAs, PTGS, RdDM, epigenetics

## Abstract

Exogenous RNA interference (exo-RNAi) is a powerful transgene-free tool in modern crop improvement and protection platforms. In exo-RNAi approaches, double-stranded RNAs (dsRNAs) or short-interfering RNAs (siRNAs) are externally applied in plants in order to selectively trigger degradation of target mRNAs. Yet, the applied dsRNAs may also trigger unintended epigenetic alterations and result in epigenetically modified plants, an issue that has not been sufficiently addressed and which merits more careful consideration.

## Exo-RNAi and Epigenetic Modifications

RNAi in plants is triggered by double-stranded (dsRNA) molecules that are cleaved by dicer-like endonucleases (DCLs) into 21–24-nt short interfering RNAs (siRNAs) [1]. More specifically, DCL4 generates 21-nt siRNAs which are loaded on argonaute 1 (AGO1) and slice complementary mRNAs in a process termed post-transcriptional gene silencing (PTGS) [2]. DCL2 produces 22-nt siRNAs which are loaded on AGO1 and either recruit RNA-directed RNA polymerase 6 (RDR6) on the complementary mRNA for the generation of secondary siRNAs [3] or repress mRNA’s translation [4]. Lastly, DCL3 processes the long dsRNA into 24-nt siRNAs that are loaded on AGO4 and are involved in RNA-directed DNA methylation (RdDM) of cognate DNA sequences [5].

DNA methylation is an important epigenetic modification and refers to the addition of a methyl group to the fifth carbon of the six-ring cytosine residue. DNA methylation was for long supposed to be induced by DNA:DNA interactions, until a breakthrough study in viroid-infected tobacco plants demonstrated that RNA:DNA interactions trigger DNA methylation, which was thus termed RNA-directed DNA methylation (RdDM) [6]. Although the exact mechanistic details of how RdDM is induced are still elusive, the current model suggests that 24-nt siRNAs dictate which DNA region is to be methylated by hybridizing either with the DNA strand or with its nascent transcript produced by RNA polymerase V (POLV) [7,8]. The interaction of 24-nt siRNA with the POLV transcript recruits, among other factors, the domains-rearranged methyltransferase 2 (DRM2) to *de novo* methylate the cytosines of the cognate DNA [9]. Although convenient, this model is not completely satisfactory. POLV seems to be recruited at already methylated DNA, thus cannot be involved in the very first step of RdDM on a completely unmethylated locus [10]. Moreover, whole-genome bisulfite sequencing revealed that RdDM is not eliminated in an *Arabidopsis* quadruple *dcl1 dcl2 dcl3 dcl4* mutant, suggesting that siRNAs (24-nt or of any other size class) are not indispensable for RdDM [11]. Indeed, DCL3 and AGO4 are not required for RdDM in inversely repeated loci that readily generate dsRNA [12,13]. It is thus more likely that the very first step of RdDM on a completely unmethylated locus is triggered not by siRNAs but by longer dsRNAs, seemingly with a minimal size of 90 bp [14,15,16]. The long dsRNA may function as a ‘ruler’ defining the DNA region that will be methylated. Whether one or both RNA strands interacts with one or both DNA strands is open to speculation, but, interestingly enough, long non-coding RNAs (lncRNAs) form triple helices with DNA to regulate gene expression, at least in mammals [17]. According to our hypothesis, the RNA:DNA interaction recruits DRM2 to establish a first (perhaps incomplete) wave of *de novo* methylation in both DNA strands (Figure 1A). To this hemi-methylated DNA, POLIV and POLV are recruited. POLIV generates short transcripts (~40 nt) that are transcribed by RDR2 into ~40-bp dsRNAs [18]. These short POL IV/RDR2 dsRNAs are processed by DCL3 into 24-nt siRNAs that are loaded on AGO4 and hybridize with POLV transcript, recruiting once more DRM2 to the hemi-methylated locus to amplify the methylation marks [19,20] (Figure 1A). Apparently, additional dsRNAs occurring from other sources (e.g., RDR6 transcription of POLII transcripts) may also contribute to this self-reinforcing amplification step [21]. When the *de novo* and amplification step are finished, all cytosines of the target DNA will be methylated in both strands and in any sequence context: CG, CHG, CHH [22,23]. Importantly, the *de novo* and amplification steps require the continuous presence of the RNA trigger (dsRNA or siRNA). In the absence of RNA trigger, CG and CHG methylation can be mitotically and meiotically maintained by methyltransferase 1 (MET1) and chromomethylase 3 (CMT3), respectively [24,25]. However, CHH methylation cannot be maintained in the absence of a RNA trigger [26] (Figure 1A).

Given the tremendous potential of RNAi to silence gene expression in almost all eukaryotes, plant biotechnologists have often resorted to RNAi tools to modify/improve crops and/or protect them against various pests and pathogens. Until recently, this was routinely achieved by transient or stable transformation of plants with transgenes designed to produce dsRNAs against the desired each time target [27,28,29,30]. However, the use of genetically modified (GM) crops has failed to gain public and political approval, hence their widespread commercialization has been rendered extremely problematic. Unsurprisingly, plant researchers have lately resorted to induction of RNAi by exogenous application of RNA molecules having the potential to trigger RNAi (exogenous RNAi, exo-RNAi), as an effective and transgene-free alternative to GM crops [31,32,33,34,35]. Indeed, dsRNAs/siRNAs were applied in plants by methods such as spraying, petiole uptake, trunk injection and root absorption in order to modify plant gene expression [36,37,38,39] and to confer resistance against viruses [40,41,42,43,44], fungi [45,46,47,48,49] and insects [50,51,52,53,54]. In the near future, one may envisage that exo-RNAi could potentially replace conventional herbicides, fungicides and insecticides, and a great amount of effort is invested in this direction by most major crop industries [55]. Of course, minimization of dsRNA production costs and optimization of dsRNA stability and uptake by the target organism is a *sine qua non* for field scale applications. To this end, several agroindustrial companies (e.g., RNAgri, agroRNA, GreenLight Biosciences) offer large amounts of dsRNA for as low as one US dollar per gram dsRNA, while various carrier compounds (e.g., clay nanosheets, chitosan nanoparticles, liposomes) were developed that significantly improve dsRNA’s stability against the environmental nucleases and uptake from the target organism. Accordingly, an exo-RNAi commercial product (‘BioDirect’ from Monsanto/Bayer) designed for insect, weed and virus control is very close to reach the market [55].

Already from the early days of RNAi discovery in plants, it has been well established that PTGS is tightly connected to RdDM [56]. It has thus been surprising that, despite the huge progress and the rapidly accumulating reports on exo-RNAi applications, the question as to whether the applied dsRNA induces not only mRNA degradation, but also DNA methylation, has skipped the attention of the researchers, with a notable exception. When Dubrovina and co-workers applied in vitro transcribed *GFP* and *NPTII* dsRNA in transgenic *Arabidopsis* carrying a *GFP/NPTII* cassette, they observed seven days post application not only *GFP* and *NPTII* mRNA downregulation, but also DNA methylation of the corresponding coding regions [38]. Arguably, transgenes are more prone to transitivity, systemic silencing and RdDM than endogenes are [57,58]. Nevertheless, native endogenes are certainly not immune to RdDM, since they can also be targeted for methylation, e.g., upon the presence of RNA viruses (exogenous RNAs themselves) that replicate through dsRNA intermediates [59,60]. Hence, the data from Dubrovina and co-workers seem to reflect a more generalized mechanism and underpin that the issue of possible epigenetic changes in exo-RNAi applications merits more careful consideration, since the plants treated with exogenous RNA may still be GM-free, but epigenetically modified, nevertheless.

Plant DCLs colocalize in the nucleus [61]. Yet, PTGS takes place in the cytoplasm [62]. Thus, during exo-RNAi, the applied dsRNA would need to first reach the nucleus and be processed by DCLs into siRNAs that would subsequently trigger PTGS in the cytoplasm. Of note, in exo-RNAi approaches that involve application of siRNAs (and not dsRNAs), transportation of siRNAs in the nucleus would not be required. Exogenous application of dsRNA in plants has been repeatedly reported to trigger PTGS of plant-encoded mRNAs [38,63,64,65,66]. This highlights that the exogenously applied dsRNA indeed manages to reach the nucleus, perhaps with the aid of dsRNA-binding proteins, where it is processed by DCLs into siRNAs. Importantly, while being in the nucleus, the exogenous dsRNA may also trigger RdDM of the cognate DNA sequences (Figure 1A). Thus, the researchers should take into consideration that when they apply dsRNA in plants to target a given mRNA for PTGS, they unintentionally also trigger RdDM of the corresponding coding region in CG, CHG and CHH context (Figure 1B). It is not clear how gene body CHG and CHH methylation affect transcription, but at least CG methylation does not seem to impede it [67,68]. Thus, in the dsRNA-treated plants, the transcript from the methylated gene body will continue to be produced, but once it reaches the cytoplasm, it will be targeted by the occurring siRNAs for degradation and PTGS (Figure 1B). It should be noted here that plants gametes are formed from somatic cells. However, even if the somatic cells from which the gametes originated were dsRNA-treated, PTGS will not be maintained in the dsRNA-free progeny and neither will CHH methylation. Yet, CG and, to lesser extent, CHG methylation will be trans-generationally inherited [68] (Figure 1B). That being said, both dsRNA-treated plants and their dsRNA-free progeny will be epigenetically modified. 

Interestingly, while both intron-containing genes and intronless genes are in principle susceptible to coding region methylation, intronless genes are much more so, presumably because their transcripts are much more prone to RDR6 processing into secondary dsRNAs and siRNAs that amplify both PTGS and RdDM. Indeed, both intron-containing and intronless genes may generate, besides the legitimate mRNAs, aberrant RNAs (abRNAs) as well, that is transcripts devoid of 5’ cap and/or 3’ polyadenylation tail. These potentially deleterious abRNAs need to be quickly eliminated by the plant cell. Thus, abRNAs from intron-containing genes are exonucleolytically degraded, due to their prior association with the spliceosome, whereas abRNAs from intronless genes are channeled to the RDR6/DCLs pathway for endonucleolytic degradation [57,58]. Of note, the transcription rate of each gene seems to be positively correlated with the likelihood for abRNA generation. Based on what was discussed above, when the exo-RNAi target is a moderately expressing intron-containing gene, then exogenous RNA application will result in local but most probably not in systemic silencing, while the risk of concomitant epigenetic modifications in the corresponding coding region will be low. In contrast, when the exo-RNAi target is a highly transcribed intronless gene, the chances to achieve both local and systemic silencing upon exogenous RNA application are high, as is also the risk of concomitant epigenetic modifications in the coding region. Conceivably, exceptions to this general rule of thumb are very likely to exist.

While DNA methylation of coding regions does not seem to affect transcription, DNA methylation of promoter regions has a very different effect. Methylated cytosines in the promoters are recognized by methyl-binding proteins (MBDs) that recruit histone methyltransferases (SUVHs) and histone deacetylases (HDACs) [69,70,71]. Histone methylation and deacetylation increase histone positive charge and thus the negatively charged DNA wraps around them more tightly in the nucleosome. As an outcome, the occurring condensed chromatin blocks POLII access and results in transcriptional gene silencing (TGS) [72,73,74,75]. It has not been tested so far whether exogenous RNAs designed to target promoter sequences can efficiently lead to TGS which, in contrast to PTGS, could be trans-generationally maintained [76] (Figure 1B). This could indeed be a challenging task, since promoter RdDM does not always lead to chromatin modifications and TGS. More specifically, while both transgenic and endogenous promoters are prone to RdDM, endogenous promoters are less prone to subsequent chromatin modifications and TGS for reasons that are not clear [77]. Interestingly, the few endogenous promoters that are prone to TGS derive from tissue-specifically expressed genes [28].

It needs to be noted here that the risk of epigenetic modifications upon exo-RNAi refers primarily to cases when the target is a plant gene. When the target is an RNA virus, then exo-RNAi will lead only to PTGS and not to RdDM, since there is no cognate DNA coding region in the virus (DNA viruses being an exception). Similarly, when the target is an insect or a fungal mRNA, then the exogenously applied dsRNA needs to function inside the insect and fungal cell, respectively, where RdDM has not been reported to occur. However, an exogenous dsRNA originally designed to function inside the insect or fungal cell, may also exhibit biological activity in the plant cell, given the opportunity. Conceivably, an exogenous dsRNA applied against insect/fungal targets will most certainly not exhibit full sequence identity to a plant target, but it may exhibit partial sequence identity with a plant locus, nevertheless. Importantly, a sequence identity as small as 30 bp between a given dsRNA and the plant DNA is enough to trigger RdDM of the plant DNA [23] (Figure 1C). Not to mention that if this 30-bp sequence identity happens to cover a plant coding region, then the 21-nt siRNAs that will be generated from the 30-bp dsRNA region will target the plant mRNA for degradation and PTGS. However, in that case, it is doubtful that the occurred cleaved transcripts will be processed by RDR6 into secondary dsRNAs and secondary siRNAs that will further amplify the silencing events in a ricochet-like manner [78]. Generation of secondary siRNAs and phased siRNAs (phasiRNAs) takes place only in few cases and only when the transcript is recognized by 21–22-nt miRNAs having an asymmetric bulge and/or by 22-nt siRNAs [3,79,80]. It was recently suggested that exogenously applied dsRNA are mainly processed to 21-nt siRNAs (and not to 22-nt siRNAs) and as such are unlikely to trigger transitivity [81]. Of note, exogenously applied 22-nt siRNAs, which are the most potent inducers of systemic silencing [37], will most certainly trigger transitive PTGS and transitive RdDM, especially in highly transcribed intronless genes.

Last, but not least, the mode of RNA application in plants greatly influences the onset of RNAi and the risk for epigenetic off-target effects. Thus, while high-pressure sprayed RNAs and/or low-pressure sprayed formulated RNAs are efficiently delivered inside the plant cells, trunk injected and petiole absorbed RNAs are transported through the apoplast and the xylem and thus do not exhibit any biologic activity in the plant cell [36,37]. Whereas symplastic RNA delivery is desired when the RNAi target is a plant gene or a virus located in the plant cell, the apoplastic RNA delivery is best suited in circumstances when the RNAi target is an insect or a fungus, since the latter need to uptake intact dsRNA (unprocessed by plant DCLs) in order to process it themselves to siRNAs with optimal biochemical RNAi properties [31]. Nevertheless, it is highly advisable that all exo-RNAi approaches, even those where the target is non-plant, are carefully designed to select non-conserved regions as targets so as to avoid undesired epigenetic off-target effects. To this end, applying unique chemically synthesized siRNAs, rather than longer dsRNAs (which are processed by DCLs into a plethora of diverse siRNA population) may also reduce off-target effects.

Overall, we strongly propose here that the occurrence of epigenetic changes in the genome upon exo-RNAi applications should be addressed and clarified in future studies. This will not only help to better interpret the obtained exo-RNAi data, but also to more comprehensively shape the regulatory framework of this exciting new technology. Although the European Food Safety Authority (EFSA) and other international risk assessment bodies and regulatory agencies have already addressed the issue of GM-based RNAi plants, there are still no clear guidelines for exo-RNAi applications [55,82].

## Figures and Tables

**Figure 1 plants-09-00673-f001:**
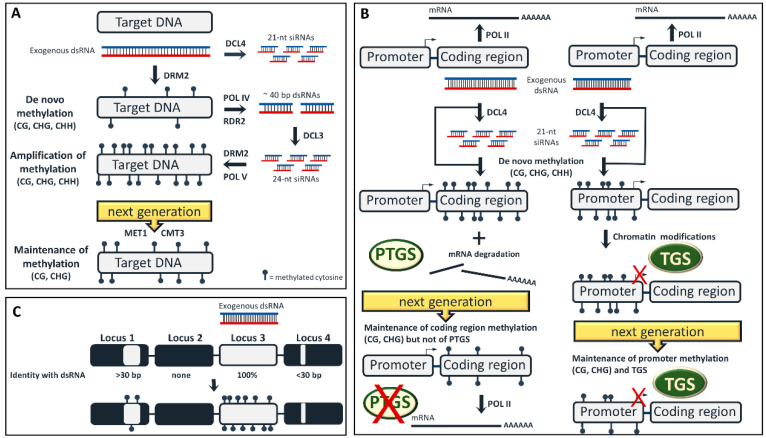
Epigenetic modifications upon exogenous RNA interference (exo-RNAi) application. (**A**) The three steps of DNA methylation: *de novo*, amplification and maintenance. Exogenously applied double-stranded RNAs (dsRNAs) are mainly processed by dicer-like 4 (DCL4) into 21-nt short-interfering RNAs (siRNAs) that have no direct role in RNA-directed DNA methylation (RdDM) in the nucleus, but they mediate post-transcriptional gene silencing (PTGS) in the cytoplasm. According to our hypothesis, the exogenous long dsRNA triggers the first incomplete wave of *de novo* methylation, while the 24-nt siRNAs, that are subsequently generated upon RNA polymerase IV (POLIV) transcription of the *de novo* methylated locus, are seemingly involved in the amplification of these methylation marks. Thus, both long dsRNAs and 24-nt siRNAs are able to recruit domains rearranged methyltransferase 2 (DRM2) to their target in a stepwise manner. In the absence of RdDM trigger molecules (dsRNA or siRNAs) CHH methylation is mitotically/meiotically lost, while CG and CHG methylation are maintained by methyltransferase 1 (MET1) and chromomethylase 3 (CMT3), respectively. (**B**) Exogenous application of dsRNA designed to target a coding region (left) will result not only to mRNA degradation aPTGS, but also to RdDM of the coding region. In the next generation, PTGS will be lost, but DNA methylation (CG and to lesser extent CHG) will be maintained. Exogenous application of dsRNA designed to target a promoter (right) will lead to promoter RdDM. Should additional chromatin modifications occur, TGS will also take place. In the next generation, CHH methylation will be lost, but CG/CHG methylation and TGS will be maintained. (**C**) A dsRNA may trigger RdDM not only to a DNA sequence which shares full sequence identity (locus 3, on-target), but also to an unrelated DNA sequence exhibiting a minimum of 30-bp sequence identity with the dsRNA (locus 1, off-target).

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
