# Peer review of "Epigenetic Modifications: An Unexplored Facet of Exogenous RNA Application in Plants"

_plants, 2020, doi:10.3390/plants9060673_

Round 1

Reviewer 1 Report

The perspective paper by Dalakouras and Papadopoulou discusses the uses of Exogenous RNA interference (exo-RNAi) as a transgene-free tool and the eventual side effects on the epigenetic state (DNA methylation) of target genes in the native genome.

The paper describes the exo-RNAi technique in a clear and simple way, with fair bibliographic support. However, I'm afraid that the "risks" of exo-RNAi associated with unintended DNA methylation of plant genome are overexploited, since, so far only one paper has reported such effect and in transgenic plants. As far as I understood the cited paper from Dubrovina et al. reported transgene methylation after exo-RNAi application of double-stranded RNA targeting the transgene. Although suggestive, it is not clear whether such effects could also be observed if a native gene would be targeted since transgene loci are prone to be targeted by methylation, due to host genome defense mechanisms (naturally occurring to control transposons).

In a specific paragraph (lines 115 to 127) authors discuss the eventual effects of exoRNAi to target promoter regions of genes and affect gene expression. In this context, I wondered how likely is it for a dsRNA to induce methylation in the promoter, if the targeted regions for exoRNAi are usually transcribed regions? How likely would it be that a dsRNA designed for a transcript (usually in low conserved regions, to avoid off-target) would target a promoter region?

Finally, in the last paragraph, the authors make bold statements, claiming that "possible epigenetic changes upon exo-RNAi need to be urgently addressed and to be more seriously taken into consideration...". I consider this a bold statement since, again, the author's claims are mostly based in one single report (Dubrovina et al) made under specific contexts (transgene-silencing).

Therefore, while I believe that the paper is suitable for publication as a perspective paper, the authors should try to adjust their claims and hypothesis to the limited observations confirming their claims. Instead of causing alarm to readers, authors should focus on proposing new ways to study these events in more detail, from a scientific perspective. Only this novel knowledge can be used to propose re-define guidelines for exo-RNAi applications. 

Minor changes:

Line 24: "trademark", consider another word

Line 26: "coined", consider another word

Line 42 to 44: consider revising to enhance clarity

Line 46: "To this...seem to be recruited", consider revising to enhance clarity

Line 102: "unintentionally", I don't think dsRNA have intentions

Line 103 to 106: consider revising to enhance clarity

Line 113: "That being said"

Line 124 to 127: consider revising to enhance clarity 

Author Response

We are thankful to the reviewers for their valuable comments to improve the quality of the manuscript. Below, please find a point-by-point response to their concerns. All changes in the manuscript are indicated with red.

Reviewer 1

The perspective paper by Dalakouras and Papadopoulou discusses the uses of Exogenous RNA interference (exo-RNAi) as a transgene-free tool and the eventual side effects on the epigenetic state (DNA methylation) of target genes in the native genome.

The paper describes the exo-RNAi technique in a clear and simple way, with fair bibliographic support. However, I'm afraid that the "risks" of exo-RNAi associated with unintended DNA methylation of plant genome are overexploited, since, so far only one paper has reported such effect and in transgenic plants. As far as I understood the cited paper from Dubrovina et al. reported transgene methylation after exo-RNAi application of double-stranded RNA targeting the transgene. Although suggestive, it is not clear whether such effects could also be observed if a native gene would be targeted since transgene loci are prone to be targeted by methylation, due to host genome defense mechanisms (naturally occurring to control transposons).

RESPONSE: It is well established that PTGS most often than not entails RdDM (a typical example being Jones et al 1999RNA-DNA interactions and DNA methylation in post-transcriptional gene silencing’). To the best of our knowledge, of all the reports on exogenous RNA application, only Dubdovina et al bothered to actually investigate the possibility of RdDM upon PTGS, and indeed concluded to the positive. Unfortunately, all other studies on exogenous RNAi simply skipped/overlooked this analysis. We certainly agree with the reviewer that transgenes are more prone to transitivity and methylation than native endogenes, but this does mean than native endogenes are immune to epigenetic modifications induced by exogenous RNAs. Indeed, plant RNA viruses (exogenous RNAs themselves) may trigger RdDM on native genes as shown in several examples, one of them being Kanazawa et al 2010 ‘Virusmediated efficient induction of epigenetic modifications of endogenous genes with phenotypic changes in plants’. We therefore believe that exogenous RNA applications may possibly lead to epigenetic modifications, at least in some cases. Nevertheless, in order to address the reviewer’s concern and to soften our message we have accordingly changed the title to ‘Epigenetic modifications: an unexplored facet of exogenous RNA application’ and revised the manuscript in several points accordingly.

In a specific paragraph (lines 115 to 127) authors discuss the eventual effects of exoRNAi to target promoter regions of genes and affect gene expression. In this context, I wondered how likely is it for a dsRNA to induce methylation in the promoter, if the targeted regions for exoRNAi are usually transcribed regions? How likely would it be that a dsRNA designed for a transcript (usually in low conserved regions, to avoid off-target) would target a promoter region?

RESPONSE: So far, the targets of exogenous RNAi are solely coding regions (thus mRNAs). In the aforementioned paragraph, we propose an approach, not tested so far, wherein promoter sequences may also be targeted by exogenous RNAs that are homologous to these promoter sequences (and not to coding regions) (Figure 1B).

Finally, in the last paragraph, the authors make bold statements, claiming that "possible epigenetic changes upon exo-RNAi need to be urgently addressed and to be more seriously taken into consideration...". I consider this a bold statement since, again, the author's claims are mostly based in one single report (Dubrovina et al) made under specific contexts (transgene-silencing).

RESPONSE: In order to soften our claims, we have changed the title and omitted strong statements such as ‘urgently’ etc. Please see the revised manuscript.

Therefore, while I believe that the paper is suitable for publication as a perspective paper, the authors should try to adjust their claims and hypothesis to the limited observations confirming their claims. Instead of causing alarm to readers, authors should focus on proposing new ways to study these events in more detail, from a scientific perspective. Only this novel knowledge can be used to propose re-define guidelines for exo-RNAi applications. 

RESPONSE: According to the reviewer’s concern, we have changed the title to ‘Epigenetic modifications: an unexplored facet of exogenous RNA application’ in order to suggest that, while the current evidence is scarce, it is nevertheless very likely that epigenetic modifications may take place and that future exogenous RNAi approaches should include this aspect in their analysis alongside the conventional mRNA downregulation assessment.

Minor changes:

Line 24: "trademark", consider another word

Line 26: "coined", consider another word

Line 42 to 44: consider revising to enhance clarity

Line 46: "To this...seem to be recruited", consider revising to enhance clarity

Line 102: "unintentionally", I don't think dsRNA have intentions

Line 103 to 106: consider revising to enhance clarity

Line 113: "That being said"

Line 124 to 127: consider revising to enhance clarity 

REPONSE: Most of these issues were revised accordingly.

Reviewer 2 Report

The paper is about the possible and overlooked epigenetic effect of exogenously applied RNA molecules. It was observed that applying exogenous RNAs not only result in PTGS of the targeted gene but its TGS as well (methylation of the coding region that is targeted). They discuss this phenomenon in great detail, although the evidence is only scarce at the moment. They warn about the significance of designing the targeting RNAs to avoid this unwanted effect.

The paper is very well written, I highly recommend publishing it, but I have a minor suggestion:

The authors forgot to discuss the monocot-specific 24-nt phasiRNA pathway in which DCL5 (a modified DCL3, actually) produce 24-nt phasiRNAs from long dsRNA templates (similarly to the 21-nt phasiRNAs produced by DCL4) during generative organ development (i.e. DOI:10.1038/s41467-019-08543-0 but there are many other papers on various monocots). The 24-nt phasiRNA pathway might be triggered by applying exogenous long dsRNAs, at least in the generative tissues or where DCL5 is expressed. It would be nice to include a short discussion of this pathway as well.

Author Response

We are thankful to the reviewers for their valuable comments to improve the quality of the manuscript. Below, please find a point-by-point response to their concerns. All changes in the manuscript are indicated with red.

Reviewer 2

The paper is about the possible and overlooked epigenetic effect of exogenously applied RNA molecules. It was observed that applying exogenous RNAs not only result in PTGS of the targeted gene but its TGS as well (methylation of the coding region that is targeted). They discuss this phenomenon in great detail, although the evidence is only scarce at the moment. They warn about the significance of designing the targeting RNAs to avoid this unwanted effect.

The paper is very well written, I highly recommend publishing it, but I have a minor suggestion:

The authors forgot to discuss the monocot-specific 24-nt phasiRNA pathway in which DCL5 (a modified DCL3, actually) produce 24-nt phasiRNAs from long dsRNA templates (similarly to the 21-nt phasiRNAs produced by DCL4) during generative organ development (i.e. DOI:10.1038/s41467-019-08543-0 but there are many other papers on various monocots). The 24-nt phasiRNA pathway might be triggered by applying exogenous long dsRNAs, at least in the generative tissues or where DCL5 is expressed. It would be nice to include a short discussion of this pathway as well.

RESPONSE: We thank the reviewer for this very insightful comment, the implications of which we have incorporated in the revised version of our manuscript. In our opinion, since the exogenous dsRNAs are exclusively processed to 21-nt siRNAs (Uslu and Wassenegger 2020), and since phasiRNAs are mainly triggered by small RNAs (siRNAs and miRNAs) having a size of 22-nt (Chen et al 2010) and/or asymmetric bulge (Manavella et al 2012) we find the possibility of phasiRNA production upon exogenous dsRNA unlikely. We hope that the reviewer will agree with us upon examining our argumentation on the revised manuscript.

Round 2

Reviewer 1 Report

Line 142 to 146. Both of these sentences are not very clear. The first sentence is not well constructed. In addition, there is the constant repetition of "very likely", "very unlikely"

Line 145 to 146. What is the reasoning behind the claim  "it is unlikely that these cleaved transcripts will be processed by RDR6" ...

Overall, authors use expressions like "it is very likely" or "it is very unlikely", or similar subjective terms to claim/suggest something. This is the type of expression that I believe to be somewhat biased. The authors can not claim that something is likely or unlikely without a clear explanation based on facts. Most of the times it seems to be based solely on their opinion. If the scientific soundness of this type of claim is not clear, most scientists will probably ignore what is being discussed.

Line 166 to 160: Consider changing to:

"Overall, we strongly propose that the occurrence of epigenetic changes in the genome upon exo-RNAi applications should be addressed and clarified in future studies. This will not only help to better interpret the obtained data but also to more comprehensively shape the regulatory framework of this exciting new technology."

Author Response

We thank the reviewer for the new suggestions, which we address below. All changes are indicated in the revised text.

Line 142 to 146. Both of these sentences are not very clear. The first sentence is not well constructed. In addition, there is the constant repetition of "very likely", "very unlikely"

RESPONSE: We have tried our best to re-write these sentences for more clarity, please see τηε revised manuscript.

Line 145 to 146. What is the reasoning behind the claim  "it is unlikely that these cleaved transcripts will be processed by RDR6" ...

RESPONSE: In plants, endogenous dsRNAs are typically processed into 21-22-24 nt siRNAs (primary siRNAs). The 21-nt primary siRNAs cleave homologous transcripts. The 22-nt primary siRNAs recruit RDR6 to their transcripts and generate additional dsRNAs that are cleaved by DCL4 into 21 nt siRNAs (secondary siRNAs). These secondary 21-nt siRNAs may further cleave homologous transcripts in a ricochet-like manner described in great detail by Voinnet et al 2008 Trends Plant Sci. Now, the exogenous dsRNAs are suggested to generate 21-nt siRNAs (and not 22-nt siRNAs, please see Uslu and Wassenegger 2020). These primary 21-nt siRNAs may result in off-target effects (transcript cleavage) but not to transitivity (secondary siRNAs) which could lead to further secondary off-targets. Overall, the reasoning behind this claim is that although exogenous dsRNAs may lead to off-targets, these  off-targets are unlikely to spread in a ricochet-like manner.

Overall, authors use expressions like "it is very likely" or "it is very unlikely", or similar subjective terms to claim/suggest something. This is the type of expression that I believe to be somewhat biased. The authors can not claim that something is likely or unlikely without a clear explanation based on facts. Most of the times it seems to be based solely on their opinion. If the scientific soundness of this type of claim is not clear, most scientists will probably ignore what is being discussed.

RESPONSE: This is a PERSPECTIVE article, and as such it is includes personal opinion. And exactly because the currently available data are scarce, the hypothesis on certain issues and the use of expressions such as likely/unlikely is unavoidable.

Line 166 to 160: Consider changing to:

"Overall, we strongly propose that the occurrence of epigenetic changes in the genome upon exo-RNAi applications should be addressed and clarified in future studies. This will not only help to better interpret the obtained data but also to more comprehensively shape the regulatory framework of this exciting new technology."

RESPONSE: We thank the reviewer for this nice suggestion, which we have incorporated in the revised manuscript.